# SCALEPERSON: TOWARDS GOOD PRACTICES IN EVALUATING PHYSICAL ADVERSARIAL ATTACKS ON PERSON DETECTION

## ABSTRACT

Person detection is widely used in safety-critical tasks but is known to be vulnerable to physical adversarial attacks. Numerous pioneering attack methods have been proposed, each claiming superior performance and exposing potential security risks. However, assessing actual progress in this field is challenging due to two common limitations in existing evaluations. First, inconsistent experimental setups and ambiguous evaluation metrics hinder fair comparisons. Second, the absence of a dedicated dataset for this task has led to evaluations on datasets originally designed for object detection, which, *while informative, are inadequate*. To address these limitations, we present a comprehensive benchmark and introduce **SCALEPERSON**, the first dataset specifically designed for evaluating physical adversarial attacks in person detection. This dataset incorporates critical factors for this task, such as person scale, orientation, number of individuals, and capture devices. Our benchmark includes standardized evaluation metrics and a modular codebase to enhance reproducibility and transparency. Leveraging this benchmark, we conduct an extensive evaluation of **11 state-of-the-art attacks** against **7 mainstream detectors** across **3 datasets**, totaling **231 experiments**. We present detailed analyses from multiple perspectives, examining the impact of various factors on the efficacy of physical adversarial attacks in person detection. The source code and dataset will be made publicly available upon acceptance of this paper.

## 1 INTRODUCTION

Person detection is a safety-critical task widely deployed in real-world applications such as autonomous driving (Chen et al., 2021) and surveillance systems (Thys et al., 2019). The reliability of these systems is paramount, as failures can lead to severe consequences, including accidents and security breaches. Consequently, extensive research has proposed physical adversarial attacks against these systems to uncover potential vulnerabilities (Huang et al., 2023). However, despite rapid developments and numerous pioneering attack methods, assessing actual progress in this field remains challenging due to two primary limitations.

Firstly, incongruent experimental setups and non-transparent implementation of evaluation metrics impede fair comparisons. For instance, many methods employ fine-tuned detectors with varying parameters. Moreover, ambiguity persists regarding whether the calculation of Average Precision (AP) employs ground truth labels or benign predicted labels as the baseline, and whether the Attack Success Rate (ASR) calculation accounts for the detector's inherent error (Huang et al., 2023). These deviations from standardization hinder fair comparisons. Secondly, there is an absence of a dedicated dataset. Existing methods are evaluated on traditional datasets like INRIAPerson (Dalal & Triggs, 2005) or COCOPerson (Lin et al., 2014), which were originally proposed for object detection tasks. These datasets only support coarse validation, precluding in-depth evaluation and analysis.

To address these limitations, we take action on two fronts: dataset and benchmark. First, we introduce SCALEPERSON, the first dataset specifically designed to evaluate physical adversarial attacks in person detection. Our SCALEPERSON dataset captures images of persons at various distances across diverse real-world scenarios, including campuses, streets, forests, and indoor settings. It provides annotations for factors critical to this task, including a person's orientation, the number of

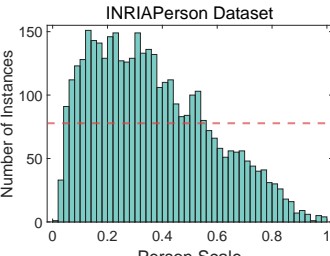

Figure 1: **Distribution of person scales in the three datasets**. Compared to the COCOPerson (Lin et al., 2014) and INRIAPerson (Dalal & Triggs, 2005), our SCALEPERSON dataset features a more uniform distribution, with an approximately equal number of samples across different scales. This facilitates a more rigorous investigation of the effect of person scale on attack effectiveness. The red dashed line indicates the mean.

persons in an image, scene type, and capture device. These annotations facilitate quantitative assessment of attack effectiveness across various perspectives, thereby establishing a more realistic and challenging testbed for evaluating physical adversarial attacks in person detection.

Notably, empirical experiments reveal that *person scale significantly impacts attack effectiveness* (see Section 2). However, existing datasets cannot support evaluation and analysis of this issue due to their imbalanced distribution of person scales (see Figure 1). Specifically, INRIAPerson has zero or single-digit person instances at certain scales, while COCOPerson, despite its sufficient sample size, contains unsuitable samples (see Figure 10). Inspired by this observation, during data collection, we deliberately controlled shooting distances to ensure a balanced distribution of person instances at each scale. Consequently, our SCALEPERSON dataset enables evaluation, comparison, and analysis of attack methods' performance across different scales.

In addition to the dataset, we present the first comprehensive benchmark of physical adversarial attack in person detection, providing a unified platform for evaluating attack effectiveness. Our modular codebase incorporates state-of-the-art (SOTA) attack methods and mainstream detection algorithms, enabling comparisons under diverse settings. The evaluation module offers a transparent protocol and implementation details, enhancing reproducibility and facilitating fair comparisons.

Leveraging these efforts, we conduct an extensive evaluation of existing attack methods under a wide range of settings, provide in-depth analysis from multiple perspectives, reveal limitations in current adversarial attacks when confronted with scale variations, and offer novel insights to inspire future advancements.

Our contributions can be summarized as follows:

- **Dataset**: We introduce SCALEPERSON, the first dataset specifically designed for evaluating physical adversarial attacks against person detection. This dataset captures images of persons at various distances across diverse real-world scenarios, addressing the uneven person scale distribution in existing datasets and enabling quantitative measurement of attack performance across different person scales.
- **Benchmark**: We develop a comprehensive benchmark for physical adversarial attacks against person detection, conducting 231 evaluations across all combinations of 11 SOTA attack methods and 7 mainstream detectors on 3 person datasets under varied settings. Our modular codebase implements a unified evaluation protocol, significantly enhancing the transparency and reproducibility of attack effectiveness assessments.
- **Analysis**: We provide multidimensional quantitative analysis, uncovering deficiencies in current methods and offering novel insights to inspire future technological advancements.

## 2 MOTIVATION

The impact of shooting distance on the performance of physical adversarial attacks against person detection is well-established (Xu et al., 2020; Wu et al., 2020; Tan et al., 2021; Huang et al., 2020; Wei et al., 2023; Zhu et al., 2022). This effect primarily stems from variations in person scale. We

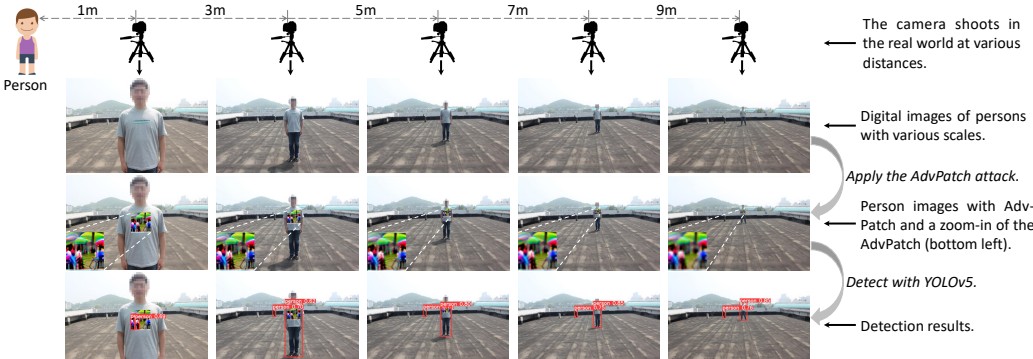

Figure 2: **A case study on the impact of person scale on attack effectiveness using the YOLOv5 detector (Ultralytics, 2020) and AdvPatch (Thys et al., 2019).** The shooting distance affects the scale of the person, subsequently influencing the size of the adversarial patch, ultimately impacting attack effectiveness.

begin by examining a practical example to explore the challenges and opportunities in evaluating such impacts.

We deploy an adversarial patch attack (Thys et al., 2019) on person images of varying scales captured from real-world scenes. As shown in Figure 2, five test images ($896 \times 596$ pixels) are used, with an initial adversarial patch size of $760 \times 760$ pixels, covering 20% of the person's height upon application. As the apply function scales the patch to maintain the 20% height ratio, significant distortion occurs, resulting in the loss of visual features (see second-row image). This leads to pronounced fluctuations in attack effectiveness when detecting images using YOLOv5 (Ultralytics, 2020) (third-row image). This example demonstrates how shooting distance affects person scale, subsequently influencing adversarial patch scaling and ultimately impacting attack effectiveness.

To quantitatively analyze attack effectiveness across different person scales, we sought a dataset suitable for comprehensive and fair evaluation. However, current datasets like COCOPerson (Lin et al., 2014) and INRIAPerson (Dalal & Triggs, 2005) are inadequate due to uneven distributions of person scales and, in COCOPerson's case, extreme occlusions of person bodies (see Figure 10). To address these limitations, we propose SCALEPERSON. As shown in Figure 1, compared to existing datasets, SCALEPERSON includes a comparable number of person instances at each scale, enabling fair quantitative assessments across various scales.

## 3 SCALEPERSON DATASET

This section introduces the new SCALEPERSON dataset, specifically designed to evaluate physical adversarial attacks against person detection. We first describe the construction process of the SCALEPERSON dataset and then analyze its statistical properties.

### 3.1 DATASET CONSTRUCTION

**Dataset Acquisition.** The primary principle of data collection was to provide a comprehensive set of person images with a uniformly distributed scale. To achieve this goal, we set up a real-world shooting setup, collecting over 5,000 person images at various scales by controlling the distance between the camera and the subject. The collected images feature diverse locations (e.g., campus, streets, forests, and indoor scenes) and were captured by a variety of imaging devices, including Sony, Canon, iPhone, Redmi, and Samsung. Additionally, we controlled the number of person instances per image, ranging from 1 to 7, to support the study of whether applying multiple adversarial perturbations to a single image results in additional attack gains. Some visual examples from our dataset are shown in Figure 3.

**Dataset Annotation.** Following most datasets (Dalal & Triggs, 2005; Lin et al., 2014) providing object-level detection, we labeled bounding boxes for person instances in the images. We utilized the `makesense.ai`[*] online toolkit for annotating the curated images. Initially, four trained annotators

---

[*]https://www.makesense.ai/

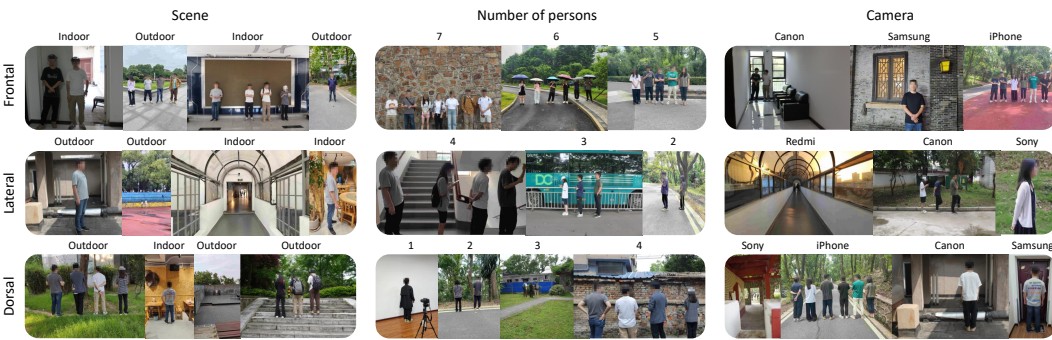

Figure 3: **The display of our SCALEPERSON dataset.** Each image is captured from various real-world scenes, including campuses, streets, forests, and indoor settings. We showcase samples across different dataset dimensions for each group of images, including person scale, orientation, number of persons, scene type, and imaging device. Leveraging the SCALEPERSON dataset, we can evaluate physical adversarial attacks against person detection from diverse perspectives.

Table 1: **Overview of existing datasets and our SCALEPERSON dataset.** We summarize whether each dataset (COCOPerson (Lin et al., 2014), INRIAPerson (Dalal & Triggs, 2005), and SCALEPERSON) considers factors that might affect the effectiveness of physical adversarial attacks.

| Dataset | Attributes of Person | | | Capture Device | Indoor/ Outdoor | Year | #Image/Person in the Dataset |
| --- | --- | --- | --- | --- | --- | --- | --- |
| | Scale | Orientation | #Person in an Image | | | | |
| COCOPerson | ✗ | ✗ | ✗ | ✗ | ✗ | 2017 | 66K/273K |
| INRIAPerson | ✗ | ✗ | ✗ | ✗ | ✗ | 2005 | 901/3890 |
| SCALEPERSON (Ours) | ✔ | ✔ | ✔ | ✔ | ✔ | - | 4K/15K |

performed the annotations based on their intuition. Subsequently, an additional reviewer identified any mislabelled samples and sent them back to the annotators for correction.

**Ethics and Privacy.** We employed 13 participants aged between 18 and 30 for data collection and annotation, compensating them at a rate of $12 per hour. The dataset construction process was conducted in strict adherence to ethical guidelines. All participants were fully informed about the nature of the research and provided explicit consent. Measures were taken to ensure no personal privacy issues were present in the dataset. The recruitment and research procedures complied with the Institutional Review Board (IRB) requirements for academic ethics.

## 3.2 DATASET STATISTICS

**Comparison with Existing Datasets.** Table 1 presents a comparative overview of popular person datasets and our proposed SCALEPERSON dataset. As observed, our SCALEPERSON dataset offers several significant advantages over existing datasets. Firstly, SCALEPERSON considers a wider variety of perspectives and richer annotations. Commonly used datasets in this field, such as IN-RIAPerson (Dalal & Triggs, 2005) and COCOPerson (Lin et al., 2014), do not account for factors such as person scale, orientation, number of persons in an image, and indoor/outdoor settings. Note that while these datasets may contain corresponding samples, they do not provide annotations for these perspectives, which SCALEPERSON does. The selected perspectives are all potential factors influencing the performance of physical adversarial attacks. Compared to typical datasets focusing only on instance-level annotations of persons, the diverse annotations in our SCALEPERSON dataset enable exploration of additional domains, such as multi-perspective attacks. This expanded research capability can facilitate broader research opportunities and promote the development of more sophisticated algorithms. Additionally, the SCALEPERSON dataset is larger than INRIAPerson but smaller than COCOPerson, balancing substantial performance with efficient computational resource usage.

**Statistical Analysis.** Figure 1 and Figure 4 present the statistical results of our SCALEPERSON dataset from several perspectives. (1) *Scale*: The scales of person instances in the SCALEPERSON dataset are uniformly distributed, facilitating fair performance comparisons of attack methods across

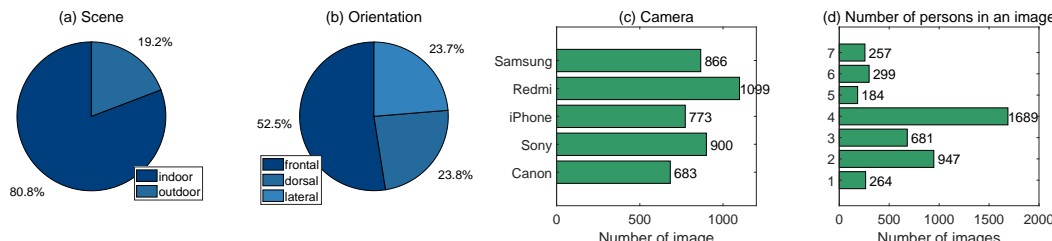

Figure 4: **Statistics of the proposed SCALEPERSON dataset**. From left to right: (a) the proportion of images in indoor and outdoor scenes, (b) the proportion of images by person orientation, (c) the proportion of images captured by different devices, and (d) the proportion of images containing varying numbers of persons.

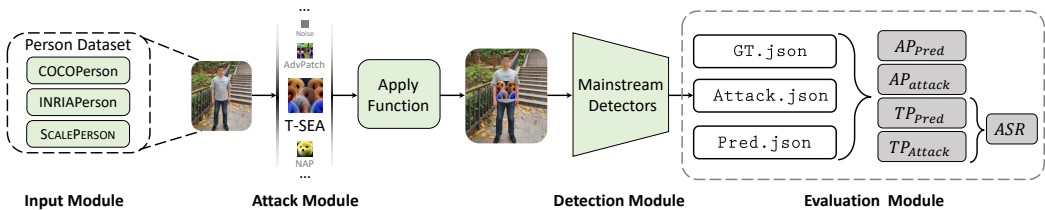

Figure 5: **The general structure of our codebase**. The unified evaluation protocol enhances the transparency and reproducibility of attack effectiveness assessments.

different scales and analysis of scale impact. (2) *Scene*: Figure 4 (a) shows the proportion of indoor and outdoor scenes in the dataset, which are 19.2% and 80.8%, respectively. (3) *Orientation*: The orientation of persons is crucial for analyzing attack methods. In the SCALEPERSON dataset, orientations are categorized as frontal, dorsal, and lateral, with proportions of 52.5%, 23.8%, and 23.7%, respectively. (4) *Camera*: We used five different devices to collect images: Canon DS126231, Sony $\alpha$6400, iPhone12, Redmi K20, and Samsung S22. Figure 4 (c) provides the distribution of these devices, allowing analysis of how different imaging devices affect attack performance. (5) *Number of persons in an image*: When multiple persons appear in an image, adversarial perturbations must be applied to each instance. Applying multiple perturbations within one image might result in interactions among the perturbations. Therefore, the SCALEPERSON dataset includes images with varying numbers of persons, as shown in Figure 4 (d).

# 4 EVALUATIONS AND ANALYSIS

## 4.1 EXPERIMENTAL SETUP

**Datasets.** We evaluate attack effectiveness on three person datasets: (1) the INRIAPerson dataset (Dalal & Triggs, 2005), a small-scale dataset containing 901 images, (2) the COCOPerson dataset (Lin et al., 2014), a subset of the MS COCO dataset that includes all images with person instances, and (3) the SCALEPERSON dataset, proposed in this paper specifically for evaluating physical adversarial attacks against person detection. Table 1 provides a summary of these datasets.

**Models.** We experiment with seven widely-used person detection models: YOLOv3 (Farhadi & Redmon, 2018), YOLOv5 (Ultralytics, 2020), YOLOv7 (Wang et al., 2023), YOLOv8 (Ultralytics, 2021), Faster R-CNN (Ren et al., 2015), Mask R-CNN (He et al., 2017), and DETR (Carion et al., 2020), all trained on the MS COCO dataset (Lin et al., 2014). For a fair evaluation, we use the official pre-trained weights for all models without any fine-tuning. The *Supplementary Material* provides download links for the corresponding weight files.

**Attack Methods.** To our knowledge, there are 11 classical methods for physical adversarial attacks against person detection in the visible modality: InvisCloak (Yang et al., 2018), AdvPatch (Thys et al., 2019), AdvT-shirt (Xu et al., 2020), UPC (Huang et al., 2020), AdvCloak (Wu et al., 2020), NAP (Hu et al., 2021), LAP (Tan et al., 2021), TC-EGA (Hu et al., 2022), T-SEA (Huang et al., 2023), AdvCaT (Hu et al., 2023), and DAP (Guesmi et al., 2024). We evaluate these methods in each setting.

Table 2: **Evaluation of attack effectiveness.** We report the Average Precision (AP%) and Attack Success Rate (ASR%) of 10 attack methods across 7 mainstream detectors on 3 datasets.

| Attack Method | YOLOv3 | | YOLOv5 | | YOLOv7 | | YOLOv8 | | Faster R-CNN | | Mask R-CNN | | DETR | |
|---|---|---|---|---|---|---|---|---|---|---|---|---|---|---|
| | AP↓ | ASR↑ | AP↓ | ASR↑ | AP↓ | ASR↑ | AP↓ | ASR↑ | AP↓ | ASR↑ | AP↓ | ASR↑ | AP↓ | ASR↑ |
| COCOPerson Dataset | | | | | | | | | | | | | | |
| Benign | 81.3 | N/A | 74.7 | N/A | 74.9 | N/A | 78.4 | N/A | 77.6 | N/A | 78.2 | N/A | 74.5 | N/A |
| Random Noise | 75.9 | 9.3 | 69.3 | 8.6 | 67.2 | 10.6 | 73.7 | 9.0 | 71.6 | 7.0 | 71.9 | 7.2 | 69.3 | 4.5 |
| InvisCloak (Yang et al., 2018) | 73.1 | 12.3 | 64.9 | 12.7 | 63.9 | 13.4 | 71.7 | 11.8 | 68.3 | 9.1 | 68.6 | 8.8 | 66.4 | 6.0 |
| AdvPatch (Thys et al., 2019) | 57.3 | 20.6 | 39.5 | 27.1 | **49.7** | 21.9 | 54.9 | 20.6 | 46.7 | 16.4 | 47.8 | 17.0 | **49.3** | **13.0** |
| AdvT-shirt (Xu et al., 2020) | 69.0 | 17.6 | 58.4 | 19.1 | 58.0 | 19.9 | 67.5 | 16.3 | 62.8 | 13.2 | 62.5 | 13.7 | 61.1 | 11.8 |
| UPC (Huang et al., 2020) | 71.4 | 11.1 | 63.6 | 11.2 | 65.9 | 10.5 | 72.7 | 9.9 | 68.9 | 6.8 | 68.8 | 8.3 | 67.4 | 4.9 |
| AdvCloak (Wu et al., 2020) | 63.5 | 18.1 | 54.5 | 19.6 | 57.8 | 18.4 | 62.7 | 17.7 | 58.1 | 12.8 | 57.8 | 15.3 | 55.6 | 11.1 |
| NAP (Hu et al., 2021) | 64.6 | 19.9 | 53.9 | 23.4 | 59.6 | 18.9 | 65.1 | 18.5 | 60.9 | 14.6 | 60.3 | 17.3 | 59.7 | 12.9 |
| LAP (Tan et al., 2021) | 72.3 | 13.4 | 62.7 | 15.3 | 62.4 | 15.0 | 70.4 | 12.8 | 66.8 | 9.6 | 67.2 | 11.3 | 65.6 | 6.7 |
| TC-EGA (Hu et al., 2022) | 69.3 | 12.4 | 65.2 | 12.3 | 63.9 | 13.5 | 71.4 | 11.3 | 68.3 | 8.7 | 66.7 | 11.1 | 66.1 | 7.3 |
| T-SEA (Huang et al., 2023) | **54.2** | **21.9** | **30.6** | **31.2** | 50.0 | **23.3** | **44.8** | **24.8** | **44.4** | **17.7** | **43.7** | **17.9** | 49.7 | 12.0 |
| AdvCaT (Hu et al., 2023) | 74.3 | 10.4 | 66.6 | 11.0 | 67.0 | 10.9 | 72.8 | 10.2 | 69.8 | 7.2 | 70.0 | 9.0 | 68.4 | 4.9 |
| DAP (Guesmi et al., 2024) | 65.4 | 18.0 | 74.7 | 18.9 | 61.6 | 16.4 | 69.0 | 14.5 | 63.9 | 11.5 | 64.1 | 13.4 | 64.3 | 8.9 |
| INRIAPerson Dataset | | | | | | | | | | | | | | |
| Benign | 88.8 | N/A | 87.8 | N/A | 86.5 | N/A | 88.3 | N/A | 88.8 | N/A | 89.4 | N/A | 86.9 | N/A |
| Random Noise | 87.2 | 2.0 | 85.9 | 3.2 | 78.2 | 8.1 | 87.2 | 2.5 | 86.5 | 1.9 | 87.7 | 1.6 | 83.7 | 1.7 |
| InvisCloak (Yang et al., 2018) | 85.4 | 3.8 | 83.5 | 4.8 | 77.8 | 7.3 | 85.5 | 3.8 | 83.8 | 5.4 | 84.1 | 4.6 | 71.5 | 7.8 |
| AdvPatch (Thys et al., 2019) | 75.2 | 9.9 | 48.4 | 22.8 | 69.9 | 13.6 | 72.6 | 13.1 | 60.0 | 10.6 | 62.5 | 11.1 | **67.4** | 10.5 |
| AdvT-shirt (Xu et al., 2020) | 84.1 | 5.1 | 73.9 | 11.9 | 78.3 | 12.8 | 83.4 | 5.5 | 79.2 | 7.9 | 80.5 | 7.6 | 78.0 | 7.5 |
| UPC (Huang et al., 2020) | 82.4 | 4.1 | 77.3 | 5.6 | 77.4 | 8.8 | 86.0 | 3.4 | 83.6 | 3.9 | 83.6 | 4.0 | 80.9 | 3.5 |
| AdvCloak (Wu et al., 2020) | 78.4 | 7.0 | 70.5 | 14.3 | 73.6 | 13.4 | 79.2 | 8.0 | 76.3 | 7.0 | 76.1 | 7.8 | 71.5 | 7.8 |
| NAP (Hu et al., 2021) | 76.4 | 8.4 | 61.2 | 16.7 | 75.1 | 11.9 | 78.2 | 9.7 | 74.9 | 7.7 | 74.9 | 8.1 | 76.6 | 7.4 |
| LAP (Tan et al., 2021) | 85.6 | 6.1 | 81.2 | 6.1 | 75.7 | 10.0 | 85.1 | 5.1 | 81.7 | 4.4 | 82.6 | 4.3 | 80.4 | 3.6 |
| TC-EGA (Hu et al., 2022) | 82.3 | 4.1 | 78.2 | 9.1 | 75.2 | 11.9 | 84.9 | 4.2 | 82.6 | 3.2 | 80.8 | 4.5 | 81.0 | 4.2 |
| T-SEA (Huang et al., 2023) | **65.4** | **13.4** | **14.8** | **36.6** | **69.4** | **13.7** | **51.7** | **20.7** | **37.9** | **20.0** | **42.0** | **19.6** | 68.2 | **11.3** |
| AdvCaT (Hu et al., 2023) | 85.3 | 3.7 | 83.5 | 3.0 | 80.2 | 5.2 | 86.8 | 2.5 | 86.4 | 1.4 | 85.8 | 2.0 | 84.0 | 1.2 |
| DAP (Guesmi et al., 2024) | 74.3 | 10.3 | 69.6 | 13.0 | 72.8 | 13.4 | 84.1 | 6.2 | 76.9 | 5.6 | 77.1 | 6.1 | 80.2 | 5.9 |
| SCALEPERSON Dataset | | | | | | | | | | | | | | |
| Benign | 96.8 | N/A | 95.5 | N/A | 96.0 | N/A | 95.9 | N/A | 97.0 | N/A | 97.0 | N/A | 96.7 | N/A |
| Random Noise | 96.6 | 0.2 | 95.4 | 0.3 | 96.0 | 0.3 | 95.8 | 0.2 | 96.8 | 0.2 | 96.7 | 0.0 | 96.2 | 0.3 |
| InvisCloak (Yang et al., 2018) | 96.5 | 0.3 | 95.1 | 0.2 | 96.0 | 0.7 | 95.9 | 0.0 | 96.7 | 0.3 | 96.5 | 0.4 | 96.3 | 0.0 |
| AdvPatch (Thys et al., 2019) | 93.9 | 0.8 | 75.5 | 6.6 | 94.0 | 0.8 | 92.3 | 1.5 | 81.4 | 1.2 | 80.2 | 1.9 | 91.1 | 0.7 |
| AdvT-shirt (Xu et al., 2020) | 95.7 | 0.9 | 93.0 | 3.1 | 95.0 | 1.1 | 95.5 | 1.0 | 93.9 | 0.8 | 93.8 | 1.2 | 95.2 | 0.8 |
| UPC (Huang et al., 2020) | 95.7 | 1.0 | 93.7 | 1.1 | 96.0 | 0.4 | 96.4 | 0.0 | 96.8 | 0.0 | 96.7 | 0.0 | 96.5 | 0.0 |
| AdvCloak (Wu et al., 2020) | 94.7 | 0.6 | 92.2 | 2.6 | 94.9 | 0.8 | 94.7 | 1.1 | 94.4 | 0.5 | 92.1 | 0.9 | 91.8 | 0.7 |
| NAP (Hu et al., 2021) | 94.5 | 1.9 | 82.4 | 6.4 | 95.0 | 1.0 | 94.6 | 0.9 | 92.3 | 0.9 | 90.0 | 1.4 | 94.8 | 0.7 |
| LAP (Tan et al., 2021) | 96.5 | 0.4 | 94.7 | 0.4 | 96.0 | 0.2 | 95.8 | 0.1 | 96.5 | 0.0 | 96.5 | 0.0 | 96.3 | 0.0 |
| TC-EGA (Hu et al., 2022) | 95.4 | 0.5 | 92.4 | 5.3 | 95.0 | 0.8 | 95.4 | 0.7 | 94.4 | 0.4 | 89.4 | 0.6 | 95.9 | 0.4 |
| T-SEA (Huang et al., 2023) | **79.9** | **5.7** | **23.0** | **17.4** | **88.7** | **2.9** | **70.7** | **5.9** | **48.5** | **4.7** | **46.7** | **6.4** | **86.2** | **1.5** |
| AdvCaT (Hu et al., 2023) | 96.4 | 0.3 | 95.1 | 0.0 | 96.0 | 0.0 | 95.9 | 0.0 | 96.7 | 0.0 | 96.7 | 0.0 | 96.7 | 0.0 |
| DAP (Guesmi et al., 2024) | 92.0 | 4.6 | 90.3 | 2.8 | 95.0 | 0.9 | 95.6 | 0.5 | 92.6 | 0.9 | 91.6 | 1.0 | 95.8 | 0.6 |

**Evaluation Metrics.** To comprehensively evaluate attack effectiveness, we employ two primary metrics: Average Precision (AP) and Attack Success Rate (ASR). AP is a widely used metric in object detection tasks, summarizing the precision-recall curve into a single value. A larger drop in AP indicates greater attack effectiveness. ASR quantifies the effectiveness of the adversarial attack by measuring the proportion of successfully attacked instances. An attack is considered successful if the adversarial perturbation causes the person detector to fail to detect the target person. ASR is calculated using the following formula:

$$ASR = \frac{TP_{\text{Pred}} - TP_{\text{Attack}}}{TP_{\text{Pred}}}, \tag{1}$$

where $TP_{\text{Pred}}$ represents the number of true positives in benign sample predictions, and $TP_{\text{Attack}}$ denotes the number of true positives in attack sample predictions. A higher ASR indicates stronger attack effectiveness.

Due to the various methods for calculating AP and the lack of provided code for ASR computation in existing approaches, our codebase implements a unified metric calculation based on the pycocotools Python package. As shown in Figure 5, the input of the evaluation module consists of three COCO-format JSON files: the ground truth labels and the prediction results with and without attacks.

**Implementation Details.** All experiments are conducted on a Linux server equipped with dual NVIDIA GeForce RTX 3090 GPUs, with all code and algorithms implemented in PyTorch. To ensure fair comparison, AP@50 is computed using the official COCO-format dataset interface, py-cocotools. The random seed is set to 7 across all experiments. The confidence and IoU thresholds are fixed at 0.25 and 0.45, respectively. For adversarial patches, we employ two configurations: A

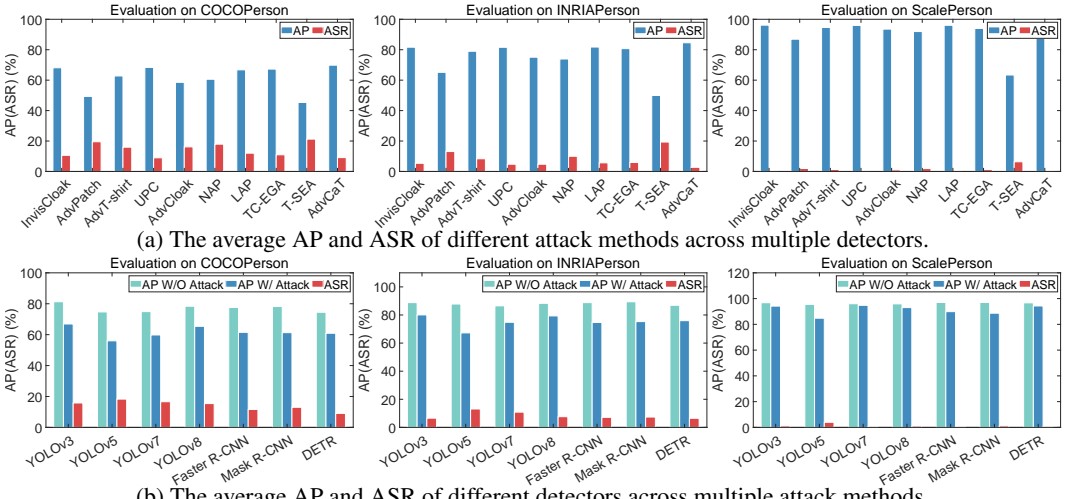

(a) The average AP and ASR of different attack methods across multiple detectors.

(b) The average AP and ASR of different detectors across multiple attack methods.

Figure 6: **Assessing the effectiveness of attack methods and the robustness of detectors.** (a) illustrates T-SEA (Huang et al., 2023) achieving the best attack effectiveness across different settings. (b) demonstrates DETR (Carion et al., 2020) detector exhibiting the least impact from attacks across different settings.

square patch with dimensions set to 20% of the person's height; A rectangular patch with width and height set to 17% and 23% of the person's height, respectively. In all cases, the adversarial patch is centrally positioned on the subject's body, both horizontally and vertically.

## 4.2 BENCHMARKING EXISTING PHYSICAL ADVERSARIAL ATTACKS

We benchmark physical adversarial attacks against person detection through extensive experiments across 3 datasets and 7 mainstream detection algorithms. Table 2 presents the Average Precision (AP%) and Attack Success Rate (ASR%). For evaluation purposes, AdvCaT was simplified to a patch-based method. We observed that while Random Noise occasionally succeeds in attacks, its effectiveness is consistently lower than all other attack methods. On the COCOPerson dataset, benign settings achieve high AP scores across all detectors, with a mean AP of 77.4%. Among attack methods, T-SEA (Huang et al., 2023) demonstrates the lowest AP and highest ASR for most detectors, notably achieving a 31.2% ASR on YOLOv5. Generally, all attack methods induce a significant drop in AP. The INRIAPerson dataset yields even higher benign AP scores, averaging 88.1%. Consistent with COCOPerson results, T-SEA exhibits strong attack performance, achieving a peak ASR of 36.6% on YOLOv5. On our SCALEPERSON dataset, benign settings achieve the highest AP scores, averaging 96.4%. This dataset demonstrates increased resilience to adversarial attacks, with many methods maintaining higher AP scores compared to other datasets. T-SEA generally exhibits the best performance across most attack settings. However, AdvPatch (Thys et al., 2019) achieves superior results in specific scenarios, such as reducing YOLOv7's AP to 49.7% on the COCOPerson.

Based on these results, we can conclude: The SCALEPERSON dataset provides a challenging benchmark, maintaining high AP scores, low ASR scores and demonstrating resilience to various attack methods. T-SEA (Huang et al., 2023) and AdvPatch (Thys et al., 2019) generally outperform other methods, indicating their effectiveness across multiple detectors and datasets. Future work should focus on developing robust attack methods tailored to varying person scales and enhancing dataset diversity to evaluate attack effectiveness better.

Additionally, Figure 6a contrasts the average AP and ASR of different attack methods across multiple detectors, revealing T-SEA (Huang et al., 2023) achieving the best attack effectiveness across different settings. Figure 6b compares the average AP and ASR of different detectors across multiple attack methods, indicating DETR (Carion et al., 2020) detector exhibiting the least impact from attacks across different settings.

Figure 7: **Attack effectiveness across person scales.** The green line represents YOLOv5 (Ultralytics, 2020) detection accuracy. Detection precision stabilizes at person scales exceeding 0.1. ASR of attack methods (AdvPatch (Thys et al., 2019) and T-SEA (Huang et al., 2023)) fluctuates, with effectiveness increasing at larger scales. In contrast, Random Noise remains consistently near zero across all scales.

Table 3: **Impact of Scene, Camera, and Number of Persons on Attack Effectiveness.** Benign precision (%) and ASR (%) for Random Noise, AdvPatch, and T-SEA are reported. All experiments utilize the YOLOv5.

| Attack Method | Scene | | Camera | | | | | Number | | | | | | |
|---|---|---|---|---|---|---|---|---|---|---|---|---|---|---|
| | indoor | outdoor | Canon | Sony | iPhone | Redmi | Samsung | 1 | 2 | 3 | 4 | 5 | 6 | 7 |
| Benign Precision | 100.0 | 94.3 | 99.3 | 99.1 | 96.1 | 86.9 | 98.9 | 92.7 | 94.2 | 99.3 | 91.6 | 99.5 | 99.7 | 99.8 |
| Random Noise | 0.0 | 0.4 | 0.0 | 0.2 | 0.8 | 1.0 | 0.0 | 0.0 | 0.6 | 0.0 | 0.6 | 0.0 | 0.3 | 0.0 |
| AdvPatch | 1.7 | 6.8 | 4.9 | 4.0 | 19.3 | 4.2 | 3.2 | 7.8 | 18.0 | 7.8 | 3.5 | 3.4 | 13.4 | 0.3 |
| T-SEA | 23.7 | 16.0 | 18.8 | 17.4 | 31.2 | 13.1 | 9.4 | 7.8 | 19.3 | 25.7 | 15.3 | 10.2 | 26.7 | 8.2 |

## 4.3 EVALUATION FROM DIVERSE PERSPECTIVES

**Person Scale.** Our proposed dataset enables the evaluation of attack effectiveness across various person scales. Figure 7 shows the ASR of Random Noise, AdvPatch (Thys et al., 2019), and T-SEA (Huang et al., 2023) across different scales. We observe that the YOLOv5 detector maintains near-perfect precision (close to 100%) once the scale exceeds 0.1. This indicates that, apart from extremely small targets, YOLOv5's detection performance is consistent across all scales, eliminating irrelevant variables in the subsequent analysis of attack fluctuations. The ASR of Random Noise consistently remains at 0. Both AdvPatch and T-SEA show fluctuations, following a similar trend: higher attack effectiveness at larger scales. This phenomenon may be attributed to extreme compression weakening the attack effectiveness of adversarial patches when applied to smaller person instances, suggesting that future attack strategies should focus on targeting smaller scales.

**Scene, Camera, Number of Persons, and Orientation.** Table 3 illustrates the impact of scene type, camera used, number of persons in an image, and person orientation. We report the benign precision (%) and ASR (%) for Random Noise, AdvPatch, and T-SEA. Our observations indicate that scene type and camera choice do not exhibit consistent patterns in their influence on attack effectiveness. Regarding the number of persons, we note a significant decrease in performance for all three attack methods when an image contains seven individuals. Figure 8 demonstrates the attack effectiveness of Random Noise, AdvPatch, and T-SEA across various person orientations. We observe that ASR is minimal for frontal attacks and maximal for lateral attacks. This finding suggests that person instances in lateral orientations are more susceptible to attacks, highlighting the need for increased focus on lateral scenarios to enhance detector adversarial robustness.

## 4.4 IMPACT OF THE DATASET IN THE TRAINING PHASE.

We designed a vanilla attack based on the PGD method (Madry et al., 2018), utilizing gradient optimization to generate adversarial patches. Using this vanilla attack, we evaluated the impact of the dataset during the training phase. As shown in Table 4, we report the metrics of adversarial patches trained on different source datasets across various settings. The results indicate that our attack method performs best when trained on the COCOPerson dataset. We speculate that the variability in person scale makes optimization challenging.

## 5 RELATED WORK

**Physical Adversarial Attacks on Person Detection.** As a safety-critical task, person detection has garnered significant interest in physical adversarial attack research. Aiming to hide persons from

Table 4: **Performance comparison of vanilla attack method trained on different datasets.** We report the AP (%) and ASR (%) of the vanilla attack trained on the three different datasets. We observe that the optimal attack effectiveness is achieved on the COCOPerson dataset.

| Source Dataset | Target Dataset | | | | | |
| | INRIAPerson | | COCOPerson | | SCALEPERSON | |
| | AP↓ | ASR↑ | AP↓ | ASR↑ | AP↓ | ASR↑ |
| INRIAPerson | 65.0 | 16.7 | 59.2 | 18.2 | 85.6 | 8.8 |
| COCOPerson | 43.2 | 49.9 | 46.3 | 36.3 | 78.9 | 31.5 |
| SCALEPERSON | 64.1 | 15.5 | 58.1 | 18.4 | 81.8 | 10.2 |

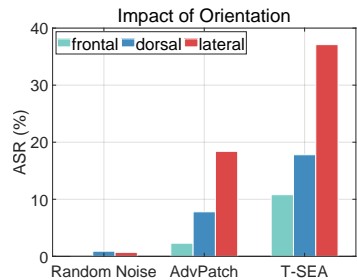

Figure 8: **Impact of person orientations**. The ASR of Random Noise, AdvPatch (Thys et al., 2019), and T-SEA (Huang et al., 2023) is shown across various orientations.

detection models, existing methods have made significant progress in terms of effectiveness (Thys et al., 2019; Wu et al., 2020; Xu et al., 2020), stealthiness (Hu et al., 2021; Tan et al., 2021), and robustness (Hu et al., 2022; Huang et al., 2023). However, some of these methods lack comprehensive evaluation metrics, and others do not provide detailed implementations for metric calculations, making it difficult to fairly compare their performance. Consequently, the actual progress in this field is difficult to assess. This paper introduces a benchmark that provides a unified setting for evaluations, ensuring consistency and reproducibility in experimental results.

**Related Benchmarks.** A benchmark in a given field can effectively evaluate progress and foster development. In the adversarial attack domain, Hingun et al. (2023) introduced a large-scale realistic adversarial patch benchmark to evaluate attacks on traffic sign recognition tasks. Zheng et al. (2023) proposed BlackboxBench, a comprehensive benchmark for black-box adversarial attacks in the digital space. Li et al. (2023) presented the Physical Attack Naturalness (PAN) dataset for benchmarking and assessing the visual naturalness of physical world adversarial attacks against vehicle detection tasks. Additionally, Li et al. (2024) introduced TA-Bench for evaluating transfer-based attacks. However, no benchmark has been established for physical adversarial attacks targeting person detection. Thus, this study endeavors to address this deficiency.

**Related Datasets.** The majority of existing physical adversarial attacks targeting person detection are trained and evaluated on the INRIAPerson dataset (Dalal & Triggs, 2005). This dataset, abundant in full-body person instances, is conducive to optimizing adversarial perturbations. Additionally, the MS COCO dataset (Lin et al., 2014) is frequently employed, which exhibits substantial variability in person representations, including instances where only partial body parts (e.g., a single hand) are visible yet still annotated as person instances (see Figure 10). These severe occlusions introduce significant challenges for the effective application of adversarial perturbations. While these datasets were originally published for training and evaluating object detectors, a dedicated dataset tailored specifically for physical adversarial attacks in person detection remains absent. To address this gap, we propose ScalePerson, a novel dataset designed to meet the unique requirements of evaluating physical adversarial attacks against person detection.

## 6 CONCLUSION

In this study, we have tackled significant limitations in assessing physical adversarial attacks on person detection systems. We introduce SCALEPERSON, the first dataset tailored for this purpose. Additionally, we establish a benchmark for evaluating attacks under diverse settings and devise a unified evaluation protocol to enhance transparency and reproducibility in assessing attack effectiveness. Leveraging these efforts, we offer quantitative analyses to provide novel insights for inspiring future technological advancements.

**Limitations**. The dataset with uniformly distributed person scales may pose challenges to attack algorithms. As discussed in Section 4.4, our vanilla PGD-based attack trained on the SCALEPERSON dataset produces adversarial patches that are less effective than those trained on the COCOPerson dataset. The greater diversity and balance of person scales necessitate optimization processes that can adapt to varying scale sizes. Future work will focus on designing attack methods that adapt to multiple scales and collaborating with experimentalists to augment the dataset.

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

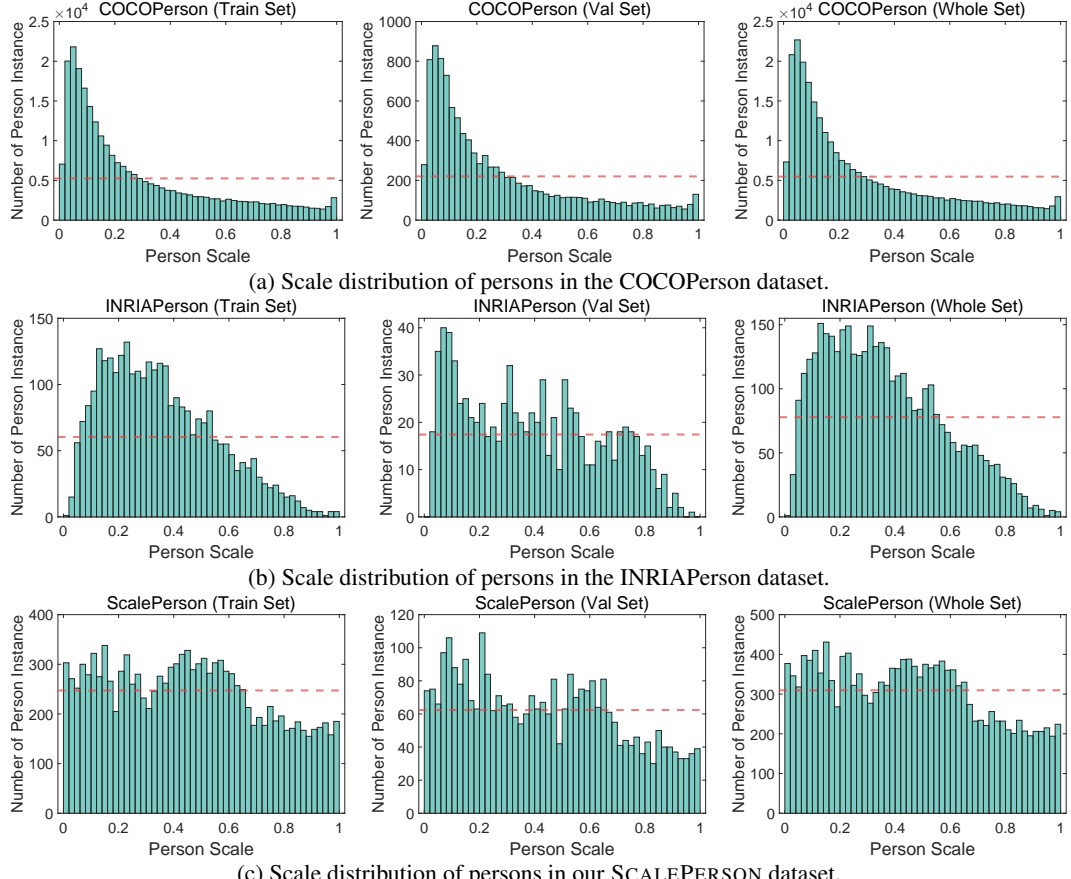

(a) Scale distribution of persons in the COCOPerson dataset.

(b) Scale distribution of persons in the INRIAPerson dataset.

(c) Scale distribution of persons in our SCALEPERSON dataset.

Figure 9: **Comparison of the distribution of scales for person instances.** COCOPerson (Lin et al., 2014) (a) and INRIAPerson (Dalal & Triggs, 2005) (b) datasets exhibit highly uneven distributions, posing challenges for testing the attack performance at different scales. Our SCALEPERSON (c) dataset offers an abundant and evenly distributed set of person instances at each scale. The red dashed line indicates the mean.

## A COMPARISON OF DATASETS IN PERSON SCALE DIMENSION

Figure 9 presents a comparison of the person scale distribution across three datasets, including the training set, validation set, and the entire dataset. We proceed with a detailed analysis.

Firstly, for the COCOPerson dataset, which is a subset of the COCO dataset (Lin et al., 2014), consisting of 66,808 images and 273,468 person instances. Its strength lies in its large and diverse data volume. However, this dataset is primarily designed to enhance the performance of object detection tasks. Many samples within it are unsuitable for training and testing physical adversarial attack methods. As shown in Figure 10, these samples contain only partial body parts of individuals, sometimes just a single organ like a hand or foot. Yet, existing physical adversarial attack methods typically apply adversarial perturbations to the central region of the body, assuming the presence of the entire person in the image. These characteristics render the COCOPerson dataset less suitable for physical adversarial attack tasks.

For the INRIAPerson dataset (Dalal & Triggs, 2005), which comprises 901 images and 3,874 person instances, most samples contain upright and unoccluded person instances, making them suitable for physical adversarial attack tasks. However, the dataset suffers from a small volume of data, particularly noticeable in the validation set, where, as seen in the visualized person scale distribution in Figure 9b, some scales have only a few samples or none at all. This limitation prevents the INRIAPerson dataset from effectively evaluating attacks at every scale.

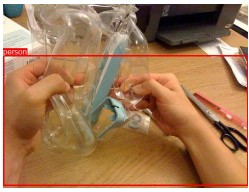 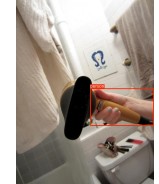 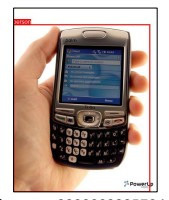 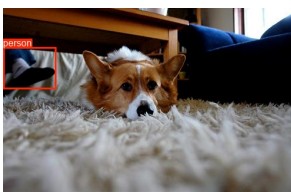

Filename: 000000251824.jpg     Filename: 000000257169.jpg    Filename: 000000332570.jpg     Filename: 000000367195.jpg

Figure 10: **Typical unsuitable samples from the COCOPerson dataset (Lin et al., 2014).** The COCOPerson dataset has the advantage of a large number of images. However, some samples are unsuitable for training and evaluating physical adversarial attacks. We present four examples with corresponding filenames, showing that these images contain only parts of a person, such as a hand or foot. In practice, many physical adversarial attack methods designed for person detection assume that the image includes the person's entire body.

In contrast, the SCALEPERSON dataset we curated consists solely of unoccluded person instances, with a deliberately controlled distribution of person scales to ensure relative uniformity. We achieved this uniformity by capturing images of persons at different distances. Our proposed SCALEPERSON dataset is specifically designed for physical adversarial attack tasks, addressing the shortcomings of the aforementioned datasets and providing a more challenging evaluation platform for this field, inspiring future technological advancements.

## B ETHICS STATEMENT

First, this work has developed a dataset and constructed a benchmark for evaluating and analyzing current methods in the domain of physical adversarial attacks on person detection systems. Our primary objective is to advance the understanding and improvement of these systems in a responsible and ethical manner.

Second, our dataset was collected in strict adherence to ethical guidelines. All participants were fully informed about the nature of the research and provided their explicit consent. Measures were taken to ensure that no personal privacy issues were present in the dataset.

## C AUTHOR STATEMENT

We, the authors of the paper titled "SCALEPERSON: Towards Good Practices in Evaluating Physical Adversarial Attacks on Person Detection", hereby declare that we bear all responsibility in case of any violations of rights or other issues related to the content of this paper. We confirm that the dataset used in our research is licensed under the Creative Commons Attribution-NonCommercial-ShareAlike (CC BY-NC-SA) license.

## D LICENSES

Table 5 provides a list of the resources that have been used in this research paper and their associated licenses. For each detector, we provide the corresponding weight files. To ensure fairness and transparency, these weight files have not undergone any fine-tuning and are directly sourced from the official repositories. The respective links are available for download.

Table 5: **Open-source resources utilized in this paper.** We list their associated licenses and links. For each detector, we provide the weight filenames with identifying information to ensure fair and transparent evaluation.

| Model | License | Weight | URL |
|---|---|---|---|
| YOLOv3 (Farhadi & Redmon, 2018) | AGPL-3.0, Enterprise | yolov3-spp.pt | link |
| YOLOv5 (Ultralytics, 2020) | AGPL-3.0, Enterprise | yolov5s.pt | link |
| YOLOv7 (Wang et al., 2023) | GPL | yolov7.pt | link |
| YOLOv8 (Ultralytics, 2021) | AGPL-3.0, Enterprise | yolov8s.pt | link |
| Faster R-CNN (Ren et al., 2015) | Apache-2.0 | faster_rcnn_r50_fpn_2x_coco.pth | link |
| Mask R-CNN (He et al., 2017) | Apache-2.0 | mask_rcnn_r50_fpn_2x_coco.pth | link |
| DETR (Carion et al., 2020) | Apache-2.0 | detr_r50_8xb2-150e_coco.pth | link |
| Pytorch | BSD-style | N/A | link |
| COCO Dataset (Lin et al., 2014) | Creative Commons Attribution 4.0 | N/A | link |
| INRIAPerson Dataset (Dalal & Triggs, 2005) | CC BY 4.0 | N/A | link |
| SCALEPERSON Dataset (Ours) | CC BY-NC-SA | N/A | N/A |

