# OpenReview forum: "ScalePerson: Towards Good Practices in Evaluating Physical Adversarial Attacks on Person Detection"
_ICLR.cc/2025/Conference — ICLR 2025 Conference Withdrawn Submission_

### Official Review · Reviewer_th75 · 2024-10-29

**Soundness:** 3
**Presentation:** 3
**Contribution:** 3
**Rating:** 6
**Confidence:** 3

**Summary:**

This work proposes a new person detection dataset, SCALEPERSON, for assessing existing physical adversarial attacking methods on the person detection tasks. It builds a standard benchmark and evaluation metrics to measure the performance of attacks under different settings, which is transparent and insightful for the future physical adversarial attacks works.

**Strengths:**

a)	This work is well organized and easy to follow. Its motivation is reasonable and provides a solid foundation for the proposed benchmark.

b)	This work conducts thorough experiments across various attacks, detectors, and datasets to construct a fair benchmark for existing methods.

c)	The quantitative analysis is detailed and uncovers weaknesses of existing datasets and methods.

**Weaknesses:**

i.	My main concern is the quality of the proposed dataset. How many unique persons are used in SCALEPERSON dataset? According to Fig 3, it seems like that the diversity of persons is low.
ii.	The AP performance is high, and ASR performance is low on the proposed dataset. Is it caused by the low difficulty and diversity of the proposed dataset? Except for T-SEA, the performance distinction of existing methods is lower on SCALEPERSON than on other datasets. Does it cause the proposed dataset not a qualified benchmark to evaluate these methods?
iii.	More statistical numbers of the proposed dataset should be provided, such as the gender ratio, occlusion levels, and ages.

**Questions:**

See weakness.

---

### Official Review · Reviewer_16Lq · 2024-11-01

**Soundness:** 2
**Presentation:** 3
**Contribution:** 1
**Rating:** 3
**Confidence:** 5

**Summary:**

The manuscript introduces SCALEPERSON, a novel dataset designed to evaluate physical adversarial attacks on person detection systems. Addressing limitations in existing evaluations—such as inconsistent setups and lack of a dedicated dataset—the paper establishes a comprehensive benchmark that standardizes evaluation metrics and includes critical factors like person scale, orientation, number of individuals, and capture devices. The benchmark assesses 11 state-of-the-art attack methods against 7 mainstream detectors across 3 datasets, totaling 231 experiments, providing detailed insights into the efficacy of these attacks.

**Strengths:**

1. Originality: The paper introduces SCALEPERSON, a novel dataset specifically designed for evaluating physical adversarial attacks on person detection systems
2. Quality: The paper features a comprehensive benchmark that systematically evaluates 11 state-of-the-art attack methods against 7 mainstream detectors on 3 datasets of person detection, ensuring robust and detailed analysis.
3. Clarity: The writing is clear and well-structured, effectively communicating the purpose and methodology behind the dataset and benchmark.
4. Significance: The introduction of SCALEPERSON advances the field by providing a resource for evaluating person detection systems.

**Weaknesses:**

1. This work focuses solely on physical attacks on person detection, which limits its generalizability and practicability, as both object detection (such as the adopted detectors) and physical attacks typically involve multiple object classes, not just persons.
2. I doubt the reasonableness of the claim that the number of persons in different scales should be evenly distributed in a dataset. Intuitively, an image can contain more small objects than large ones, so an even distribution of objects across various scales could lead to an imbalance in the number of images with different object sizes. This raises the question of which factor is more significant. Moreover, natural images often include objects in significantly different scales, which raises concerns about the reasonableness of using the introduced ScalePerson for evaluating attack performance on other physical dynamics besides scale.
3. Physical factors are not well aligned in data collection, which may lead to misleading experimental results and conclusions, as previous works have demonstrated that some physical dynamics can also be exploited to perform attacks.
4. Physical attacks should be conducted in real-world scenarios, whereas the perturbations are applied in the digital domain in the experiments. How, then, do the results demonstrate physical attack performance?

**Questions:**

Please refer to the weaknesses.

---

### Official Review · Reviewer_cqsS · 2024-11-03

**Soundness:** 2
**Presentation:** 2
**Contribution:** 2
**Rating:** 5
**Confidence:** 3

**Summary:**

This paper addresses the problem of evaluating physical adversarial attacks on person detection systems. The main issues highlighted are the lack of consistent experimental setups and ambiguous evaluation metrics that hinder fair comparisons, and the absence of a dedicated dataset designed for assessing physical adversarial attacks, leading to evaluations on datasets not ideally suited for this purpose.

The authors propose SCALEPERSON, the first dataset specifically designed for evaluating physical adversarial attacks in person detection. This dataset incorporates critical factors such as person scale, orientation, number of individuals, and capture devices, providing a more realistic and challenging testbed for evaluating such attacks. Additionally, they introduce a comprehensive benchmark with standardized evaluation metrics and a modular codebase to enhance reproducibility and transparency.

**Strengths:**

1. SCALEPERSON is the first dataset designed to address the uneven distribution of person scales in existing datasets, which is crucial for evaluating the effectiveness of adversarial attacks across different scales.
2. The benchmark includes standardized evaluation metrics and a modular codebase that allows for transparent and reproducible assessments of attack effectiveness.
3.  The authors conduct an extensive evaluation of 11 state-of-the-art attacks against 7 mainstream detectors across 3 datasets, providing multidimensional quantitative analysis.
4. The analysis uncovers deficiencies in current methods and offers novel insights to inspire future technological advancements.

**Weaknesses:**

1. While SCALEPERSON addresses the issue of uneven person scale distribution, it may not cover all possible real-world scenarios, which could limit the generalizability of the findings. The collection and use of images in the dataset must adhere to strict ethical guidelines to ensure personal privacy is not compromised.
2. The effectiveness of the benchmark relies on the selection of attack methods included. If certain effective attacks are not considered, the benchmark may not fully represent the threat landscape.

**Questions:**

Pls see the weaknesses above

---

### Official Review · Reviewer_vanB · 2024-11-04

**Soundness:** 3
**Presentation:** 4
**Contribution:** 2
**Rating:** 5
**Confidence:** 4

**Summary:**

This work introduces a novel dataset and benchmark for physical adversarial attacks on person detection task, focusing on fair comparison regarding various factors such as scale, orientation, cameras, etc. Also, this work suggests an evaluation metrics: Average Precision (AP) and Attack Success Rate (ASR) for benchmark. With the dataset and benchmark, the authors conduct an extensive evaluation with various attack methods and detectors across the existing and novel datasets.

**Strengths:**

1. This work provides a novel dataset designed for studying physical adversarial attack. The dataset consists of person images with an uniformly distributed scale, while the existing datasets (INRIAPerson, COCOPerson) do not.
2. The presentation is good.
3. This work provides the extensive experimental results comparing the various adversarial attack methods between datasets.

**Weaknesses:**

1. In the SCALEPERSON dataset, the Average Precision (AP) for both benign and attacked settings appears to be too high, with small variance in scores across methods, except for AdvPatch and T-SEA. In other words, the proposed dataset seems too easy (to detect person), lacking the discriminative power needed to serve as an effective benchmark. The dataset is supposed to contain more dynamic scenes.

2. As shown in Table 3, the influence of scene type varies across different attack methods. Therefore, to enable a fair comparison, the proportion of indoor and outdoor scene images is supposed be more balanced, as is the case with the distribution of camera types.

3. The advantage of using Attack Success Rate (ASR) as a metric is not clearly explained, for example, in comparison to Average Precision (AP).

4. The ASR metric only accounts for detector false negatives (FNs, missed detections) caused by physical adversarial attacks and does not consider detector false positives (FPs). However, physical adversarial attacks also appear to cause detection FPs, as shown in Figure 2.

**Questions:**

Please refer to the weakness part.

---

### Note · Authors · 2024-11-13

I have read and agree with the venue's withdrawal policy on behalf of myself and my co-authors.